New austrolimulid from Russia supports role of Early Triassic horseshoe crabs as opportunistic taxa

Bicknell Russell D.C. 1 rdcbicknell@gmail.com
http://orcid.org/0000-0003-4508-9259 Shcherbakov Dmitry E. 2
1 Palaeoscience Research Centre, School of Environmental and Rural Science, University of New England , Armidale, NSW , Australia
2 Borissiak Paleontological Institute, Russian Academy of Sciences , Moscow , Russia
Lieberman Bruce
Electronic publication date: 2021 Jun 30
Publication date: 2021
Volume: 9
Electronic Location ID: e11709
Received 2021 Apr 13; Accepted 2021 Jun 9
Copyright: © 2021 Bicknell and Shcherbakov
Copyright year: 2021
Copyright holder: Bicknell and Shcherbakov
License: This is an open access article distributed under the terms of the Creative Commons Attribution License, which permits unrestricted use, distribution, reproduction and adaptation in any medium and for any purpose provided that it is properly attributed. For attribution, the original author(s), title, publication source (PeerJ) and either DOI or URL of the article must be cited.
License URL: https://creativecommons.org/licenses/by/4.0/

Keywords: Xiphosurida, End-Permian extinction, Triassic recovery, Geometric morphometrics, New species, Exceptional preservation

Funding: University of New England Postdoctoral Research Fellowship Russian Science Foundation 21-14-00284 This research was supported by funding from a University of New England Postdoctoral Research Fellowship (to Russell D.C. Bicknell) and by the Russian Science Foundation grant 21-14-00284 (to Dmitry E. Shcherbakov). The funders had no role in study design, data collection and analysis, decision to publish, or preparation of the manuscript.

==============================
Horseshoe crabs are extant marine euchelicerates that have a fossil record extending well into the Palaeozoic. Extreme xiphosurid morphologies arose during this evolutionary history. These forms often reflected the occupation of freshwater or marginal conditions. This is particularly the case for Austrolimulidae—a xiphosurid family that has recently been subject to thorough taxonomic examination. Expanding the austrolimulid record, we present new material from the Olenekian-aged Petropavlovka Formation in European Russia and assign this material to Attenborolimulus superspinosus gen. et sp. nov. A geometric morphometric analysis of 23 horseshoe crab genera illustrates that the new taxon is distinct from limulid and paleolimulid morphologies, supporting the assignment within Austrolimulidae. In considering Triassic austrolimulids, we suggest that the hypertrophy or reduction in exoskeletal sections illustrate how species within the family evolved as opportunistic taxa after the end-Permian extinction.

Introduction

Examining ecological recovery from the “mother of all extinctions” (the end-Permian extinction) during the Triassic is important for understanding how biological systems can redevelop after major devastating events (Erwin, Bowring & Yugan, 2002; Jablonski, 2002; Payne et al., 2004; Twitchett et al., 2004; Dineen, Fraiser & Sheehan, 2014). Triassic vertebrate (Hu et al., 2011; Chen & Benton, 2012; Benton et al., 2013; Tintori et al., 2014; Fu et al., 2016), invertebrate (Rodland & Bottjer, 2001; Hu et al., 2011; Chen & Benton, 2012; Hofmann et al., 2013; Fu et al., 2016; Ponomarenko, 2016), and trace (Chen, Fraiser & Bolton, 2012; Crasquin & Forel, 2014; Luo & Chen, 2014; Luo et al., 2019, 2020; Shi et al., 2019; Xing et al., 2021) fossil assemblages have been examined to understand recovery of the distinct palaeoecological facets. The arthropod record in particular has shed light on how marine and terrestrial groups recovered after the end-Permian. Ostracods (Crasquin-Soleau et al., 2007; Forel, 2012; Forel et al., 2013; Crasquin & Forel, 2014; Chu et al., 2015) and insects (Gall & Grauvogel-Stamm, 2005; Shcherbakov, 2008a, 2008b; Hu et al., 2011; Żyła et al., 2013; Ponomarenko, 2016; Zheng et al., 2018) are commonly examined, with rarer studies of branchiopods (Żyła et al., 2013) and horseshoe crabs (Gall & Grauvogel-Stamm, 2005; Hu et al., 2011; Lerner, Lucas & Lockley, 2017; Bicknell et al., 2019b; Bicknell, Hecker & Heyng, 2021). The record of Triassic xiphosurids (so-called horseshoe crabs) has recently been scrutinised, a research trajectory that has uncovered a wealth of data on post-Permian taxa (see Błażejowski et al., 2017; Hu et al., 2017; Lerner, Lucas & Lockley, 2017; Bicknell et al., 2019a, 2019b, 2021; Bicknell & Pates, 2020; Lamsdell, 2020). Two xiphosurid families are known from the Triassic: Austrolimulidae and Limulidae (Table 1). Of these two, austrolimulids are predominantly marginal marine to freshwater forms that commonly exhibit hypertrophied or reduced features. Here, we present new horseshoe crab material from the Konservat Lagerstätte within the Petropavlovka Formation, Cis-Urals of Russia to promote the study of Austrolimulidae and their role in the Triassic recovery of Xiphosurida. This material is also examined using geometric morphometrics to mathematically illustrate the austrolimulid position of these fossils within xiphosurid morphospace. This evidence, coupled with a thorough taxonomic consideration, prompted us to place the Petropavlovka Formation material within a novel genus and species: Attenborolimulus superspinosus gen. et sp. nov.

Table 1 Summary of known Triassic xiphosurids.

Taxon	Family	Formation, locality	Age	Depositional environment	
Austrolimulus fletcheri Riek, 1955	Austrolimulidae	Hawkesbury Sandstone, New South Wales, Australia	Middle Triassic (Anisian, 247.2–242 Ma)	Marginal marine to freshwater	
Attenborolimulus superspinosus gen. et sp. nov.	Austrolimulidae	Petropavlovka Formation, Cis-Urals, Russia	Early Triassic (Olenekian, 251.2–247.2 Ma)	Marginal marine to freshwater	
Batracholimulus fuchsbergensis (Hauschke & Wilde, 1987)	Austrolimulidae	Exter Formation, Germany	Late Triassic (Rhaetian, 208.5–201.3 Ma)	Marginal marine to freshwater	
Dubbolimulus peetae Pickett, 1984	Austrolimulidae	Ballimore Formation, New South Wales, Australia	Middle Triassic (Ladinian)	Marginal marine to freshwater	
Limulitella bronni (Schimper, 1853)	?Austrolimulidae	Grés á Voltzia Formation, France	Middle Triassic (Anisian)	Marginal marine to freshwater	
Limulitella liasokeuperinus (Braun, 1860)	?Austrolimulidae	?Exter Formation–?Bayreuth Formation, Germany	Late Triassic-Early Jurassic (?Rhaetian- Hettangian, 208.6–199.3 Ma)	Marginal marine to freshwater	
Limulitella tejraensis Błażejowski et al., 2017	?Austrolimulidae	Ouled Chebbi Formation, Tunisia	Middle Triassic (Anisian-Early Ladinian, 247.2–237 Ma)	Marginal marine to freshwater	
Limulitella volgensis Ponomarenko, 1985	?Austrolimulidae	Rybinsk Formation, Russia	Early Triassic (Olenekian)	Marine	
Psammolimulus gottingensis Lange, 1923	Austrolimulidae	Solling Formation, Germany	Early Triassic (Olenekian, Spathian, 251.2–247.2 Ma)	Marginal marine to freshwater	
Vaderlimulus tricki Lerner, Lucas & Lockley, 2017	Austrolimulidae	Thaynes Group, Idaho, USA	Early Triassic (Olenekian, Spathian)	Marginal marine	
Heterolimulus gadeai (Vía & De Villalta, 1966)	Limulidae	Alcover Limestone Formation, Spain	Middle Triassic (Ladinian)	Marine	
Keuperlimulis vicensis (Bleicher, 1897)	Limulidae	Marnes Irisées Supérieures Formation, France	Late Triassic	Marine	
Mesolimulus crespelli Vía Boada, 1987	Limulidae	Alcover Limestone Formation, Spain	Middle Triassic (Ladinian)	Marine	
Sloveniolimulus rudkini Bicknell et al., 2019b	Limulidae	Strelovec Formation, Slovenia	Middle Triassic (Anisian)	Marine	
Tarracolimulus rieki Romero & Vía Boada, 1977	Limulidae	Alcover Limestone Formation, Spain	Middle Triassic (Ladinian)	Marine	
Yunnanolimulus (?) henkeli (von Fritsch, 1906)	Limulidae	Jena Formation, Germany	Middle Triassic (Anisian)	Marine	
Yunnanolimulus luopingensis Zhang et al., 2009	Limulidae	Guanling Formation, Luoping, China	Middle Triassic (Anisian)	Marine	
Note:

Taxa are order by family and then alphabetically by genus and species. Temporal data taken from Bicknell & Pates (2020), Bicknell, Naugolnykh & Brougham (2020), Bicknell et al. (2021) and Bicknell, Hecker & Heyng (2021)). Note the uncertain placement of Limulitella in Austrolimulidae, and Yunnanolimulus henkeli. In Fig. 10 and Supplemental Information 3, Limulitella is placed within Limulidae, and Yunnanolimulus (?) henkeli is referred to Limulitella henkeli (following Bicknell et al., 2021).

Geological history and setting

The Permian–Triassic succession of the Cis-Urals is well known for diverse fossil tetrapods and stratigraphic sections that permit detailed study of changes in climate, landscapes, vegetation, and biological communities across the Permian–Triassic boundary (Ochev & Shishkin, 1989; Shishkin et al., 1995; Benton, Tverdokhlebov & Surkov, 2004; Gomankov, 2005; Shcherbakov, 2008a; Benton & Newell, 2014; Shishkin & Novikov, 2017). The Petropavlovka Formation within this important succession is considered upper Olenekian (251.2–247.2 Ma) based on the Parotosuchus Otschev & Shishkin (in Kalandadze et al., 1968) tetrapod fauna, the lungfish Ceratodus multicristatus (Vorobyeva & Minikh, 1968), miospore assemblages rich in Densoisporites nejburgii associated with the lycophyte Pleuromeia, and magnetostratigraphy (Fig. 1A; Shishkin et al., 1995; Minikh & Minikh, 1997; Tverdokhlebov et al., 2003). During the Olenekian, orogenic development occurred in the Ural Mountains, while the Peri-Caspian Depression was inundated by the transgression of the Palaeotethys. This resulted in increased rates of siliciclastic deposition in the Cis-Urals (Tverdokhlebov, 1987). In the Cis-Ural Trough and southeastern slope of the Volga-Ural Anteclise, a vast lacustrine-deltaic floodplain was formed. This bordered the northern Peri-Caspian marine basin of the Palaeotethys. The Petropavlovka Formation accumulated in this floodplain. The formation consists of grey and reddish-grey siliciclastics. It is primarily a rhythmic alternation of coarse- and fine-grained sandstone, clay, siltstone, and fine-grained clayey sandstone, reaching a total thickness of 400–800 m (Shishkin et al., 1995). Conglomerate lenses are also common. Coarser sediment represents alluvial deposits, while finer lithologies constitute shallow water lacustrine deposits. These facies characterise the delta floodplain and delta front complexes that comprise the Petropavlovka Formation (Tverdokhlebov et al., 2003).

Figure 1 Geographical and geological information for the studied fossil site.

(A) Map showing locality of Petropavlovka III (red star). Dotted line represents boundaries of tectonic regions, modified from Shcherbakov, Vinn & Zhuravlev (2021). (B) Stratigraphic log of Petropavlovka II–IV sections showing location of horseshoe crab-bearing lens (modified from Tverdokhlebov, 1967).

The Petropavlovka Formation stratotype section occurs along the Sakmara River and adjacent ravines close to Petropavlovka ~45 km north-east of Orenburg (52°02′ N, 55°38′ E). Red beds exposed here yield tetrapods, lungfish, clam shrimps (conchostracans), and ostracods (Shishkin et al., 1995). Along one ravine, a one-meter-thick lens of grey fine-grained polymictic siltstone to sandstone was identified (locality Petropavlovka III, bed 43; Tverdokhlebov, 1967). The lens contains abundant plant megafossils including sphenophytes and gymnosperms (Dobruskina, 1994). In 2018–2019 numerous diverse insects wings, millipedes, horseshoe crabs, microconchids, and a microdrile oligochaete were collected in the lens, along with seed fern pinnules and lycophyte fragments (Hannibal & Shcherbakov, 2019; Shcherbakov et al., 2020; Shcherbakov, Vinn & Zhuravlev, 2021).

Materials and Methods

The studied specimens were collected by the field parties of, and are housed in, the Borissiak Paleontological Institute (PIN), Russian Academy of Sciences, Moscow, Russia. The material was photographed with a Nikon D800 camera mounted with a Nikon AF-S ED Micro Nikkor 60 mm f/2.8G lens. Images were z-stacked with Helicon Focus Pro 6.7. Furthermore, a Leica DFC425 camera coupled to Leica M165C stereomicroscope was used. Finally, to examine possible evidence for finer structures, specimens were examined under a TESCAN VEGA scanning electron microscope (SEM) housed at the PIN. A backscattered electron detector was used as the specimens were not coated.

When describing the material, we followed the systematic taxonomy of Bicknell & Pates (2020) and Bicknell et al. (2021) and used anatomical terms presented in Lerner, Lucas & Lockley (2017), Bicknell (2019), Bicknell, Naugolnykh & Brougham (2020), and Bicknell et al. (2021).

The geometric morphometric analysis presented here develops on recent applications by Bicknell (2019), Bicknell & Pates (2019), Bicknell et al. (2019b), and Lustri, Laibl & Bicknell (2021). The approach was used to assess where the Petropavlovka Formation material falls in xiphosurid morphospace and allows for a mathematical comparison with other xiphosurid specimens, augmenting the taxonomic description presented here. A total of 103 specimens arrayed across 23 genera from Austrolimulidae, Limulidae, and Paleolimulidae (sensu Bicknell & Pates, 2020) were considered. Landmarking and semilandmarking was conducted with the Thin-Plate Spline (TPS) suite (http://life.bio.sunysb.edu/morph/index.html). A TPS file was constructed using tpsUtil64 (v.1.7). The TPS file was imported into tpsDig2 (v.2.26). This software was used to place four landmarks on the right prosomal section, as well as 50 semi-landmarks along the right prosomal shield border (Fig. 2; Table S1). Points were digitised as xy coordinates. When the right side was poorly preserved, the left side was used, and data mirrored. These points populated the TPS file with landmark data (Supplemental Information 1). The TPS file was imported into R. The ‘geomorph’ package (Adams, Collyer & Kaliontzopoulou, 2020) was used to conduct a Procrustes Superimposition and Principal Components Analysis (PCA) of the data (Supplemental Information 2). Only the first two Principal Components (PCs) were considered as they explain 75.6% of the variation in the data (Supplemental Information 3). Bicknell et al. (2019b) demonstrated that the distribution in PC space reflects biological variation. As such, while preservational mode varies between specimens (consider Bicknell & Pates, 2020), this variation has little impact on the morphospace (see discussion in Lustri, Laibl & Bicknell, 2021). The generic and family assignments presented in Supplemental Information 3 reflect a combination of taxonomic theses presented in Bicknell & Pates (2020), Lamsdell (2020), and Bicknell et al. (2021).

Figure 2 Depiction of geometric morphometric data gathered here: four landmarks and one semilandmark outline.

Consider Table S1 for description of landmarks.

The electronic version of this article in Portable Document Format (PDF) will represent a published work according to the International Commission on Zoological Nomenclature (ICZN), and hence the new names contained in the electronic version are effectively published under that Code from the electronic edition alone. This published work and the nomenclatural acts it contains have been registered in ZooBank, the online registration system for the ICZN. The ZooBank LSIDs (Life Science Identifiers) can be resolved and the associated information viewed through any standard web browser by appending the LSID to the prefix http://zoobank.org/. The LSID for this publication is: 5435A6BA-AE34-4698-8872-7A350DB799B1. The online version of this work is archived and available from the following digital repositories: PeerJ, PubMed Central and CLOCKSS.

Systematic Palaeontology

Family Austrolimulidae Riek, 1955

Genus Attenborolimulus gen. nov.

Etymology: The generic name is given in honour of Sir David Attenborough and his unparalleled contributions to natural history and conservation. His last name is combined with Limulus—the most-well documented extant xiphosurid genus.

Type species: Attenborolimulus superspinosus, new species.

Diagnosis. Austrolimulid with anteriorly effaced, ridge-less cardiac lobe, slightly splayed genal spines extending posteriorly to three-fourths of thoracetron length with occipital bands extending to spine terminus, tubercle structures along posterior prosomal and anterior thoracetronic border, medial thoracetronic lobe lacking a sagittal ridge, and long, strongly keeled telson.

Attenborolimulus superspinosus sp. nov.

Figures 3–8

Figure 3 Holotype of Attenborolimulus superspinosus gen. et sp. nov. PIN 5640/220, counterpart.

(A and B): Photograph and interpretative drawing. Abbreviations: Car: cardiac lobe; Eye: lateral compound eye; Fla: thoracetronic flange; Fs: fixed spine; Med: medial thoracetronic lobe; Oph: ophthalmic ridge; Pa: prosomal appendage; Pro: prosoma; Thr: thoracetron; Tel: telson. Image credit: Sergey Bagirov.

Figure 4 Holotype of Attenborolimulus superspinosus gen. et sp. nov., PIN 5640/220, part.

(A and B): Photograph and interpretative drawing. Abbreviations: Car: cardiac lobe; Fla: thoracetronic flange; Med: medial thoracetronic lobe; Oph: ophthalmic ridge; Pa: prosomal appendage; Pro: prosoma; Thr: thoracetron; Tel: telson. Image credit: Dmitry Shcherbakov.

Figure 5 SEM images of the Attenborolimulus superspinosus gen. et sp. nov.

(A, C and D): Holotype, PIN 5640/220, counterpart. (A) Entire specimen. (C) Close up of box in (A), showing small moveable spine notches and fixed spines (white arrows). (D): Close up of box in (A), showing tubercles along prosomal thoracetronic border (white arrows). (B): Paratype, PIN 5640/200, part. Image credit: Dmitry Shcherbakov.

Figure 6 Paratype PIN 5640/200 of Attenborolimulus superspinosus gen. et sp. nov. showing key prosomal features.

(A and B): Part, photograph and interpretative drawing. (C and D): Counterpart, photograph and interpretative drawing. Abbreviation: Car: cardiac lobe. Image credit: (A) Sergey Bagirov; (C) Dmitry Shcherbakov.

Figure 7 Paratype PIN 5640/217 of Attenborolimulus superspinosus gen. et sp. nov.

(A and B): Photograph and interpretative drawing. Image credit: Sergey Bagirov.

Figure 8 Reconstruction of Attenborolimulus superspinosus gen. et sp. nov.

Reconstruction credited to Katrina Kenny.

Etymology: Species name reflects the hypertrophied (super-) genal spine (-spinosus) morphology.

Holotype: PIN 5640/220 (part and counterpart).

Paratypes: PIN 5640/217, PIN 5640/200 (part and counterpart).

Type locality and horizon. Petropavlovka III near the village of Petropavlovka, Orenburg region, Russia; Petropavlovka Formation, upper Olenekian, Lower Triassic.

Diagnosis. Same as for genus.

Preservation. Specimens are preserved as partly domed exoskeletons as part and counterpart on yellowish or grey siltstone.

Description. PIN 5640/220 (part and counterpart): An articulated prosoma, thoracetron, and distally incomplete telson (Figs. 3–5). Specimen is 32.0 mm long as preserved. Prosoma parabolic in outline, 9.8 mm long at midline, and 15.3 mm wide between genal spine tips. Exoskeletal warping along anterior and left lateral prosomal sections. Prosomal rim 0.2 mm wide. Prosomal doublure 1.6 mm wide laterally, arcuately widened to 2.5 mm medially. Ophthalmic ridges curved towards the lateral prosomal border, ~4.5 mm long. Ridges do not converge anteriorly. Lateral compound eyes narrow and reniform, ~2.9 mm long, ~0.7 mm wide, inner orbita 4.1 mm from midline. Cardiac lobe 7.5 mm long, 4.1 mm wide posteriorly, tapering to its mid-length, about 2.0 mm wide in anterior half, tapered to 1.4 mm near apex, effaced anteriorly. Break in left genal spine within first quarter of thoracetron. Posterior-most left genal section 8.4 mm from midline. Angle between inner edge of left genal spine and left thoracetron side 77.2°. Right genal spine complete, terminates three fourths along thoracetron. Genal spine terminus 7.8 mm from midline, 6.9 mm from level of prosomal-thoracetronic hinge. Angle between right inner edge of genal spine and right thoracetron side 38.5°. Pronounced occipital bands extend from ophthalmic ridges to genal spine ends. Prosomal-thoracetronic hinge pronounced, 7.6 mm wide, and 0.6 mm long. Posterior prosomal border with shallow central notch 2.1 mm wide. Distal sections of prosomal appendages noted lateral to compound eyes (Fig. 3B).

Thoracetron trapezoidal, completely preserved in counterpart (Figs. 3 and 5C), 8.1 mm long at midline, 9.4 mm wide anteriorly, tapering to 4.7 mm posteriorly. Tubercle structures along anterior thoracetron border noted under SEM (Fig. 5D). Thoracetronic flange present. Rounded anterolateral lobes apparently present. Setose margins of branchial appendages (opercula) visible anteriorly on left side in counterpart. Medial thoracetronic lobe weakly defined, 7.3 mm long, 3.0 mm anteriorly, tapering to 1.2 mm posteriorly. Lobe lacking medial thoracetronic ridge. Left pleural lobe has 0.3 mm wide rim. Left lobe 8.0 mm long, 2.6 mm wide, tapering posteriorly to short, round terminal spine. Right lobe damaged in part. Measurements taken from counterpart. Right lobe 8.2 mm long, 2.5 mm wide, tapering posteriorly to short, rounded terminal spine. Minute fixed spines and movable spine notches observed under SEM on left side of thoracetron (Fig. 5C). Telson 14.1 mm long as preserved, with well-developed keel. Telson terminates at rock edge, has a kink at a third of the spine length.

PIN 5640/200 (part and counterpart): Isolated prosoma preserved more completely in part (Figs. 5B and 6). Prosoma parabolic in outline, 15.1 mm long at midline, and 28.0 mm wide between most distal genal spine points. Exoskeletal warping along anterior and right lateral prosomal sections. Prosomal rim 0.6 mm wide. Prosomal doublure 2.1 mm wide, arcuately widened backwards up to 4.1 mm medially. Right ophthalmic ridge noted in counterpart (Figs. 6C and 6D). Ridge curved towards the lateral prosomal border, 9.1 mm long. Lateral compound eyes narrow reniform, ~3.7 mm long, ~0.8 mm wide, right inner orbita 7.5 mm from midline. Cardiac lobe present, 7.5 mm long, 6.8 mm wide posteriorly, tapering (posteriorly to anteriorly) to 1.8 mm, effaced anteriorly. Left genal spine broken distally. Most distal left genal section 13.9 mm from midline. Right genal spine complete, lateral margin slightly convex. Genal spine terminus 14.1 mm from midline, 13.6 mm from prosomal-thoracetronic hinge. Pronounced occipital bands extend from ophthalmic ridges to genal spine ends, better preserved along right genal spine. Ridge delimiting occipital band with tubercles along posterior prosomal border and near base of genal spines. Posterior prosomal border with arcuate central notch 4.3 mm wide. Clam shrimp (round structures) noted.

PIN 5640/217: Central and left side of prosoma (Fig. 7), 15.4 mm long at midline, and 17.1 mm wide at widest section. Prosomal rim 0.3 mm wide. Partial left ophthalmic ridge noted. Cardiac lobe 9.0 mm long, 7.0 mm wide posteriorly, tapering slightly anteriorly to 2.5 mm, effaced anteriorly. Anterior most section of left genal spine noted. Two tentaculitoid tubeworms noted on left side of prosoma (round structures; Shcherbakov, Vinn & Zhuravlev, 2021).

Remarks: The horseshoe crab material documented herein displays hypertrophied genal spines, a feature common in Belinurina and Austrolimulidae. The examined material lacks the expression of thoracetronic tergites extending from the medial lobe to the thoracetron edge and a rounded thoracetron common to Belinurina. This suggests the material likely belong within Austrolimulidae. Bicknell, Naugolnykh & Brougham (2020) outlined two major groupings of austrolimulids: those with reduced thoracetronic sections relative to the prosoma and those with genal spines that extend up to the thoracetron terminus. Prosomal and thoracetronic sections of the Petropavlovka Formation specimens are comparable, excluding this material from the first group Bicknell, Naugolnykh & Brougham (2020) outlined. This differentiates the material considered here from Batracholimulus fuchsbergensis (Hauschke & Wilde, 1987), Boeotiaspis longispinus (Schram, 1979), Dubbolimulus peetae Pickett, 1984, Panduralimulus babcocki Allen & Feldmann, 2005, and Shpineviolimulus jakovlevi (Glushenko & Ivanov, 1961). Comparisons to Austrolimulus fletcheri Riek, 1955, Franconiolimulus pochankei Bicknell, Hecker & Heyng, 2021, Psammolimulus gottingensis Lange, 1923, Tasmaniolimulus patersoni Bicknell, 2019, and Vaderlimulus tricki Lerner, Lucas & Lockley, 2017 are therefore needed, as they are austrolimulids with hypertrophied genal spines. Austrolimulus fletcheri and V. tricki both have hypertrophied genal spines with extensive splay, which is not observed in the Petropavlovka Formation material (Riek, 1955, 1968; Lerner, Lucas & Lockley, 2017). Franconiolimulus pochankei, the youngest austrolimulid, has a cardiac ridge, distally effaced occipital bands, and a thoracetronic free lobe, none of which are observed in the Petropavlovka Formation material. Tasmaniolimulus patersoni has pronounced thoracetronic free lobes, as well as keeled cardiac and medial thoracetronic lobes (Ewington, Clarke & Banks, 1989; Bicknell, 2019). These are not observed in the Petropavlovka Formation material, excluding the new fossils from this Lopingian (259.1–251.9 Ma) genus. Psammolimulus gottingensis is the most morphologically similar to the new material. Indeed, the genal spine morphology and pronounced occipital bands suggest a strong alignment with P. gottingensis (Meischner, 1962). However, P. gottingensis has hypertrophied terminal thoracetronic spines and pronounced free lobes. Neither of these features are observed in the specimens examined here. Based on this comparison, we assert that the Petropavlovka Formation material is morphologically distinct from other austrolimulids enough to be separated at the generic level, as Attenborolimulus superspinosus gen. et sp. nov. This taxonomic assessment is supported by geometric morphometric results (see “Results”).

One point to consider is Limulitella Størmer, 1952 as an austrolimulid genus. Lamsdell (2020) recently used tree topology to propose that Limulitella fell into Austrolimulidae, suggesting that the family consisted of limuloids with “apodemal pits present on thoracetron; thoracetron lacking tergopleural fixed spines; posteriormost thoracetron tergopleurae swept back and elongated to form ‘swallowtail’; axis of thoracetron bearing dorsal keel” (Lamsdell, 2020, p. 20). Examining L. bronnii (Schimper, 1853), for example, specimens have evidence of fixed spines, rendering the placement of Limulitella within Austrolimulidae tenuous. This perspective is supported by the position of Limulitella within morphospace that has consistently been closer to members of Limulidae of Paleolimulidae than Austrolimulidae (Bicknell, 2019; Bicknell et al., 2019b; Bicknell & Pates, 2019; Figs. 9 and 10). It therefore seems more likely that Limulitella species represent a group of limulids, rather than austrolimulids (sensu Bicknell & Pates, 2020; Bicknell et al., 2021). At best, Limulitella may represent a transitional form between the two families. Finally, comparing the morphologies of Limulitella presented in Bicknell & Pates (2020, figs. 28–30) to our material, the lack of hypertrophied genal spines separates this genus from Attenborolimulus superspinosus.

Figure 9 Three examined xiphosurid families in PC space.

Austrolimulids occupy most positive PC1 space while limulids and paleolimulids are mostly constrained to negative PC1 space. Attenborolimulus superspinosus gen. et sp. nov. falls within the convex hull occupied by Austrolimulidae. Note that the austrolimulid morphospace excludes Limulitella specimens, as the position of this genus in Austrolimulidae is considered dubious.

Figure 10 PC plot showing morphospace occupied by xiphosurid genera.

Where more than one specimen of the same genus was digitised, genera are bound by convex hulls. Attenborolimulus superspinosus gen. et sp. nov. is not bound by any convex hull, excluding the specimen from other genera.

Results

The PCA plot illustrates that PC1 (48.3% shape variation) describes how laterally the most distal genal spine point extends from the sagittal line (Fig. 9). PC2 (27.3% shape variation) describes how posteriorly the genal spine projects, relative to the prosomal sagittal line and posterior border. Paleolimulids and limulids are both located in PC1 space <0.05, reflecting the lack of genal splay observed in the groups. Specimens within Austrolimulidae cover PC1 space from 0–0.3 reflecting the variation in genal spine splay observed in the family. The holotype of Attenborolimulus superspinosus gen. et sp. nov. is located in a positive PC1 space (PC1 = 0.099) and a neutral PC2 space (PC2 = 0.002) (Figs. 9 and 10). It therefore falls outside the morphospace occupied by Limulidae and Paleolimulidae (Fig. 9) and within the morphospace occupied by Austrolimulidae. Furthermore, it is distinct from the distribution of other austrolimulid genera (Fig. 10).

Discussion

The meter-thick lens that yielded Attenborolimulus gen. nov. is a rare occurrence of grey lithologies among the red beds of the Petropavlovka Formation. The red beds yield the lungfish Ceratodus, temnospondyl amphibians, and procolophonid and erythrosuchid reptiles (Shishkin et al., 1995; Novikov, 2018). By comparison, the grey lens contain a different set of fossils: abundant, but fragmentary vascular plants, numerous insects (mainly isolated wings of various roaches, beetles, hemipterans, and rare dragonflies, grylloblattids, and orthopterans), microconchids, rare millipedes, and a microdrile oligochaete (Hannibal & Shcherbakov, 2019; Shcherbakov et al., 2020; Shcherbakov, Vinn & Zhuravlev, 2021). Clam shrimp and ostracods recorded in the grey bed occur in surrounding red beds as well. Notably, plant and animal fossils are not restricted to certain bedding planes but are randomly distributed in the rock, thus preserving some three-dimensionality. Such sediment probably accumulated in an ephemeral pond during a flood event. The millipedes, most plants, and nearly all insects were washed into the water body from the land and are therefore allochthonous fossils. The horsetails Equisetites and Neocalamites likely grew as helophytes protruding out of the water as some fragments of their stems are encrusted with microconchid shells. The aquatic ecosystem is represented by (sub)autochthonous fossils of ceratodontid lungfishes, numerous schizophoroid beetle adults, clam shrimp, ostracods, horseshoe crabs, microdriles, and microconchids. The microdrile specimens represent the earliest fossil record of oligochaete annelids. This small worm is similar to modern tubificids, and its relatively well-developed body wall musculature suggests that sediment burrowing was originally another way to escape desiccation on the bottom of seasonally drying ponds (Shcherbakov et al., 2020). Minute microconchids that encrusted plant stems, horseshoe crab exuvia, and other available firm substrates represent the major suspension feeders in the Petropavlovka ecosystem. These extinct lophophorates were genuine disaster taxa—eurytopic stress-tolerators that flourished in the aftermath of the end-Permian extinction in both marine and continental basins all over the world (Shcherbakov, Vinn & Zhuravlev, 2021). Dense accumulations of primarily pyrite dodecahedra are common on the plant stem fragments and attached microconchid tubes. A high carbon/sulphur ratio might have produce abundant pyrite clusters in a freshwater basin (Hethke et al., 2013). Also, the decomposition of organic matter by sulphate-reducing bacteria favoured increased acidity and would lead to the precipitation of early diagenetic pyrite (Fürsich & Pan, 2016). This sedimentological feature might be indicative of abundant decaying plant and animal remains consumed by bacteria at the lake bottom, but not for the redox state of the water column itself. However, a lacustrine palaeocoenosis, including ceratodontid lungfishes capable of aestivation in their burrows, horseshoe crabs, microdriles, and abundant microconchids, strongly supports a meromictic eutrophic lake.

Vacant ecological space is a key factor in allowing evolutionary innovation to develop (Erwin, 2008). Triassic austrolimulids capitalised on vacant marginal to freshwater environs left after the end-Permian extinction, thus exploiting an unprecedented array of niches and representing ‘disaster forms’ (sensu Schubert & Bottjer, 1992). Triassic forms exhibit more extreme morphologies than their Late Paleozoic counterparts (e.g., Panduralimulus babcocki, Shpineviolimulus jakovlevi, and Tasmaniolimulus patersoni) suggesting that the morphological stock required for Triassic diversification had arisen prior to the end-Permian (Bicknell, 2019). The high xiphosurid Triassic diversity and disparity, followed by a constrained morphology and generic level diversity from the Jurassic, records the extinction of austrolimulids (Bicknell, Hecker & Heyng, 2021) and the transition to a morphology that was conserved through into modern ecosystems (Bicknell & Pates, 2020). The hypertrophied genal spines observed in austrolimulids also illustrate evolutionary convergence with the Pennsylvanian-aged belinurids Euproops Meek, 1867 and Belinurus Pictet, 1846. The prevalence of this trait in two distinct xiphosurid families demonstrates how colonisation of marginal conditions placed similar evolutionary constraints on the xiphosurid body plan, resulting in comparable morphologies.

Supplemental Information

Supplemental Information 1 R code.

Click here for additional data file.

Supplemental Information 2 TPS file of analysed specimens.

Click here for additional data file.

Supplemental Information 3 Summary of digitised landmarks depicted in Figure 2.

Click here for additional data file.

Supplemental Information 4 Data used for semilandmark sliding.

Click here for additional data file.

Supplemental Information 5 PCA results.

Includes family, generic and temporal data.

Click here for additional data file.

We are deeply grateful to Anastasia Felker, Elena Lukashevich, Olesya Strelnikova, Maria Tarasenkova, Alexey Bashkuev and Dmitry Vasilenko who participated in fossil collecting, and especially to Eugeny Karasev who found the holotype, to Sergey Bagirov for the excellent photographs, to Roman Rakitov for his help in obtaining perfect SEM images, and to Andrey Sennikov (all PIN) and Valentin Tverdokhlebov (Saratov State University) for information on the fossil locality. We thank Nicolas E. Campione for discussions around the topic of disaster taxa and Katrina Kenny for her exceptional reconstruction of Attenborolimulus superspinosus. Finally, we thank Jason Dunlop, Joachim Haug, and Thomas Hegna for their comments that thoroughly improved the text.

Additional Information and Declarations

Competing Interests

Author Contributions

Data Availability

New Species Registration

The authors declare that they have no competing interests.

Russell D.C. Bicknell conceived and designed the experiments, performed the experiments, analyzed the data, prepared figures and/or tables, authored or reviewed drafts of the paper, and approved the final draft.

Dmitry E. Shcherbakov conceived and designed the experiments, performed the experiments, analyzed the data, prepared figures and/or tables, authored or reviewed drafts of the paper, and approved the final draft.

The following information was supplied regarding data availability:

The raw morphometric data, the data needed to perform the analysis, and the PCA data are available in the Supplementary Files.

Specimens are reposited in the Borissiak Paleontological Institute (PIN), Russian Academy of Sciences, Moscow, Russia:

PIN 5640/220 part and counterpart

PIN 5640/200 part and counterpart

PIN 5640/217

The following information was supplied regarding the registration of a newly described species:

Publication LSI: urn:lsid:zoobank.org:pub:5435A6BA-AE34-4698-8872-7A350DB799B1.

Genus name: urn:lsid:zoobank.org:act:11531F97-2ACA-411F-8002-E4E3426C22B6.

Species name: urn:lsid:zoobank.org:act:8FF20D49-096E-451E-9A06-284DC1665B1A.

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
