# Peer review of "New austrolimulid from Russia supports role of Early Triassic horseshoe crabs as opportunistic taxa"

_PeerJ, doi:10.7717/peerj.11709_

## Round 0.1 · original submission · Minor Revisions

The reviewers have some minor suggested revisions that will be useful to make. Please take a look at those and revise accordingly. When you resubmit, please provide a letter documenting the changes you have made in response to the reviewers as well as a version of the manuscript that shows the changes tracked. Look forward to seeing your revision.

·

Basic reporting

The basic paper is very good. I would have liked to see more phylogenetic context for Attenborolimulus (i.e. the discussion of Limulitella).

Experimental design

The morphometric analysis is a little bit of pattern searching without a hypothesis driving it. I think this could be fixed with a little reframing of the analysis.

Validity of the findings

If you are going to talk about austolimulids diversifying in the wake of the end Permian extinction, you might briefly consider their origins--do they originate in the late Paleozoic? Early Mesozoic? Do they just *pop* into existence in the early Mesozoic?

Additional comments

Pay close attention to the comments in the reviewed pdf.

·

Basic reporting

No comment, all OK.

Experimental design

All OK

Validity of the findings

All OK

Additional comments

This manuscript describes a new species of fossil horseshoe crab and places it in its wider context as part of a putative radiation of taxa after the Permian mass extinction even. Principal component analysis is also applied to help resolve its systematic placements. The manuscript is well written, the descriptions of the new species are of a high standard and the figures are very good. I think the manuscript is a useful contribution to the systematics and evolution of Mesozoic horseshoe crabs and is acceptable for publication subject to MINOR REVISION as outlined below.

TITLE

I’m not sure “uncovers” is the correct word here. A single species alone is not really evidence for all horseshoe crabs being disaster taxa. Maybe “…supports the role of Triassic horseshoe crabs…” would be better?

ABSTRACT

line 21: “…these bizarre forms illustrate that…”


INTRODUCTION

line 33: “…recovery of distinct the palaeoecological…” Do you mean “…of the distinct…”.


MATERIALS AND METHODS

line 92: “A backscattered electron detector…“

line 95: “…presented in Lerner et al., (2017), Bicknell (2019)…etc.”

line 99: “…material falls in morphospace…”


SYSTEMATIC PALAEONTOLOGY

lines 139/143: style question? Should these sentences end in a full stop?

line 197: What are “clam shrimps”, I’m not familiar with this term so maybe you need to add the scientific name of the group.

line 2017: “Furthermore, as Belinurina went extinct by the end-Permian…” maybe qualify this by adding “on current data” or something like that.?There is always the chance that a Triassic belanurid could be found later in which case age alone would not exclude this taxon.

line 230: “B Based on this…” [delete “B”?]


RESULTS

line 245: “The PCA plot…”


REFERENCES

line 377: should “Wein“ be “Wien“ (for Vienna?).

line 481: italicise “Tarracolimulus rieke“

line 481: Is the species name correct? In Table 1 (p. 33) it is listed as „rieki”?


FIGURES

line 555: there seems to be part of the legend missing in the text here. Later with the figures we have the information, i.e.

“(A, B): Photograph and interpretative drawing. Image credit: Sergey Bagirov. Abbreviations: Car: cardiac lobe; Eye: lateral compound eye; Fla: thoracetronic flange; Fs: fixed spine; Med: medial thoracetronic lobe; Oph: ophthalmic ridge; Pa: prosomal appendage; Pro: prosoma; Thr: thoracetron; Tel: telson.”

line 573: “Three examined xiphosurid families in…“

·

Basic reporting

Overall the story is straight forward, expanding the diversity of Triassic representatives of Xiphosurida. My two main suggestions are rather simple: 1) Avoid Linnean ranks wherever possible. You can only do it wrong, or better, there is no right way as ranks are a matter of opinion and nothing else. 2) Throw out the 'disaster taxa'. I do not have the impression that this is a well developed concept, but just the attempt to push another buzz word.

Hence I suggest to accept the paper after minor revision.

Experimental design

no comment

Validity of the findings

no comment

Additional comments

General: Linnean ranks are subjective categories. There is no objective criterion when which rank should be applied. I can therefore only suggest to not use these categories.

Title: This is not a geologists' journal, most readers (including me) will have to look up what “Olenekian” is. “Disaster taxa” is not a well known concept, in fact I am not sure whether it is indeed a well formulated one. I suggest to avoid it altogether and use an easily comprehensible phrasing.
Throughout the manuscript the geological ages need to be accompanied by rough estimations of age for non-geologists.

Abstract: The first sentence includes many general statements of which I am not sure if they are correct. Why are they archetypical, the majority of extant forms of Chelicerata sensu stricto living in the marine realm (more than 1000 species of sea spiders) look quite different. In which way is the fossil record exceptional? Compared to which other lineages?

As pointed out I would prefer to see no use of Linnean ranks. Yet, using such ranks demands for following specific rules. For example, the expression “xiphosurid” as used here is ambiguous. In most lineages the ending '-id' is derived from the family name, hence '-idae' (as it is done in other instances in the text). Yet here it is, as far as I get it, derived from Xiphosurida. Hence it should correctly be derived as “xiphosuridan”.

Line 129: Especially here the presented ranks are highly subjective, and many authors would disagree on the here assigned ranks. Just leave them out.

Line 155: not entirely sure, but I guess that this is not the prosomal shield

Line 159: if the anterior tagma is called prosoma, why then use here a malacostracan term (cephalothorax)? This is in my view inconsistent.

Line 205: what is 'complete expression of tergites'? Does this refer to them not being articulated? Please express in simple words.

Line 278: I do not get the concept of disaster taxa. It seems not a widespread known term. Also as it is explained here I personally would think they are more something like “post-disaster taxa”. Also the term “taxon” makes the expression strange as it might well read as referring to a single species. I do not see why the story gets better with this term.

Line 302: Maybe it is because I am a German native speaker, but Bauplan is a very unfortunate term in these days. Since Intelligent Design is around, we need to avoid any reference to a planer of evolution.

---

## Round 0.2 · accepted · Accept

The authors have done a good job addressing the comments from the previous round of review. The paper is ready to move forward to the next stage of production.